# Induction of Drug-Resistance and Production of a Culture Medium Able to Induce Drug-Resistance in Vinblastine Untreated Murine Myeloma Cells

**DOI:** 10.3390/molecules28052051

**Published:** 2023-02-22

**Authors:** Valentina Laghezza Masci, Davide Stefanoni, Angelo D’Alessandro, Marta Zambelli, Lorenzo Modesti, Daniele Pollini, Elisa Ovidi, Antonio Tiezzi

**Affiliations:** 1Laboratory of Plant Cytology and Biotechnology of Natural Compounds, Department for the Innovation in Biological, Agri-Food and Forestal Systems, University of Tuscia, Largo dell’Università, snc, 01100 Viterbo, Italy; 2Department of Biochemistry and Molecular Genetics, University of Colorado Anschutz Medical Campus, Aurora, CO 80045, USA; 3Department of Medical Sciences, Laboratory for Technologies of Advanced Therapies (LTTA), University of Ferrara, 44121 Ferrara, Italy; 4Department of Cellular, Computational and Integrative Biology, University of Trento, 38123 Trento, Italy

**Keywords:** vinblastine, drug resistance, vinblastine-resistant cells, metabolomic analyses, murine myeloma cells

## Abstract

Cancer therapies use different compounds of synthetic and natural origin. However, despite some positive results, relapses are common, as standard chemotherapy regimens are not fully capable of completely eradicating cancer stem cells. While vinblastine is a common chemotherapeutic agent in the treatment of blood cancers, the development of vinblastine resistance is often observed. Here, we performed cell biology and metabolomics studies to investigate the mechanisms of vinblastine resistance in P3X63Ag8.653 murine myeloma cells. Treatment with low doses of vinblastine in cell media led to the selection of vinblastine-resistant cells and the acquisition of such resistance in previously untreated, murine myeloma cells in culture. To determine the mechanistic basis of this observation, we performed metabolomic analyses of resistant cells and resistant drug-induced cells in a steady state, or incubation with stable isotope-labeled tracers, namely, ^13^C ^15^N-amino acids. Taken together, these results indicate that altered amino acid uptake and metabolism could contribute to the acquisition of vinblastine resistance in blood cancer cells. These results will be useful for further research on human cell models.

## 1. Introduction

Current pharmaceutical treatments in anticancer therapies use different compounds of synthetic and natural origin. However, despite some positive results, relapses are common, as standard chemotherapy regimens are not fully capable of completely eradicating cancer stem cells [1]. The phenomenon of drug resistance is a process that confers cancer (stem) cells resistance and tolerance to anticancer agents. The mechanism of action of drug resistance is complex and influenced by several factors, such as (i) direct drug inactivation [2,3]; (ii) the alteration of the drug target [4]; (iii) drug efflux [5], a mechanism that is dependent upon mitochondrial ATP levels, which fuel the ATP cassette (ABC) transporter-dependent systems that are responsible for pumping the drug out of the cell [6]; (iv) activation of the DNA damage repair mechanisms [7]; (v) inhibition of cell death [8]; (vi) intrinsic cellular heterogeneity [9]; (vii) metabolically controlled epigenetic effects [10,11,12]; (viii) or any combination of these mechanisms [13,14].

Of all the mechanisms described above, the failure of chemotherapies is most commonly attributed to two main causes: multidrug resistance (MDR) [15,16], caused by an overexpression of drug efflux pumps, a family of transporter protein membranes such as the ABC transporters mentioned above [6], and the existence of tumor activating cells or cancer stem cells (CSCs) [17], which—especially in blood cancers—are usually characterized by unique metabolic adaptations (e.g., reliance on mitochondrial metabolism as opposed to a Warburg-like, glycolytic phenotype of highly proliferating cells) [18]. As such, understanding the mechanism of metabolic adaptation underlying the development of drug resistance could inform novel or orthogonal therapeutic strategies in clinics.

A wide range of natural products (NPs) derived from plants, marine organisms and microorganisms have been introduced into medical treatments and used to optimize the development of new potent and tolerated drugs [19]. These molecules, naturally produced and used by organisms for defensive or adaptive purposes, have been used as healing agents for thousands of years, and still today continue to be the most important source of new potential therapeutic preparations [20]; in addition, the increasing interest in and the large number of NPs presently investigated in laboratories all over the world seem to open new perspectives for new and accurate pharmaceuticals [21,22,23,24,25]. In a landmark paper in 2020, Newman and Cragg updated and expanded their previous review on the importance of NPs-based compounds as drug candidates in anticancer therapies, and presented a broad overview of approved anticancer agents, emphasizing that 80% were of natural origins and the remainder purely synthetic [26]. Four classes of NPs are currently used in clinical application for anticancer treatments: vinca alkaloids, epipodophyllotoxin lignans, diterpenoid taxanes and the quinoline alkaloid derivatives of camptothecin [27]. Vinblastine [28,29], vincristine [30], podophyllotoxin [31], paclitaxel [32] and camptothecin [33] are the most effective natural anticancer compounds, and can be considered as “pioneer chemistry” for the development of anticancer agents [34,35]. Such compounds, similar to many other natural products, can regulate immune function, inhibit cell proliferation, and even induce cell death by apoptosis, autophagy or ferroptosis [36]. For example, vinblastine has been originally tested in clinics as a potential therapeutic in the treatment of multiple myeloma [37], and later became a mainstay in the treatment of other blood cancers (e.g., Hodgkin’s lymphoma) [38]. Unfortunately, resistance to vinblastine is observed in the clinical treatment of patients with multiple myeloma, though the mechanism driving the acquisition of such resistance is incompletely understood [39,40].

In order to improve the knowledge of drug resistance related to vinblastine use, we report here the results of our investigations based on the treatment of P3X63Ag8.653 murine myeloma cells by incubation with low concentrations of vinblastine in culture media [41]. This treatment has led to the selection of vinblastine-resistant cells; in addition, we cultured previously untreated murine myeloma cells with the culture media of selected vinblastine-resistant cells, and the induction of drug-resistance occurred. Given the role of metabolic reprogramming in the acquisition of drug resistance (e.g., via the fueling of MDR transporters, such as the P-glycoprotein p, Pgp became involved in resistance to vinblastine) [42], here, we have performed metabolomic analyses of resistant cells and drug resistant-induced cells at steady states, or incubation with stable isotope-labeled tracers, namely, ^13^C ^15^N-amino acids. The obtained results clearly show the existence of differences at the metabolic level among the untreated cells (control cells), resistant cells and drug resistant-induced cells. Our results will inform further investigations aimed at testing the impact of vinblastine on human cancer cell lines and primary tumors.

## 2. Results

### 2.1. Induction of Resistance and MTT Assay

Considering the MTT assay as a reliable tool for measuring cell viability and proliferation, the assays were performed to check for drug resistance in resistant cells (Res) and induced cells (Ind), and the results obtained are shown in Figure 1. Before proceeding to detect lower sensitivity or resistance to vinblastine treatment by the P3X63Ag8.53 cells, MTT assays were performed to identify the treatment concentration that led to the loss of viability of half of the cells (EC_50_) [43,44]. The treatment for 24 h with a concentration of 20 nM of vinblastine, capable of reducing cell proliferation by 50%, showed 58.19 ± 1.92% of cell viability in P3X63Ag8.53 (Ctrl+), while in the cells maintained under the same growth conditions as resistant cells (Ctrl Res) and induced cells (Ctrl Ind), 56.94 ± 2.61% and 58.64 ± 2.74% of cell viability were detected, respectively. Res cells or those that received treatment for 7 days with a vinblastine dosage equal to 1/4 of the EC_50_ value (5 nM) showed a significant increase in cell viability of 86.09 ± 3.08%. Similarly, Ind cells, i.e., those grown for 24 h in culture medium from the resistant cells (IM24), but not for 7 days with low dosages of vinblastine, showed 92.69 ± 2.09% cell viability, thus confirming that drug resistance was induced in P3X63Ag8653 cells.

### 2.2. Metabolomics Analyses Highlight a Significant Up-Regulation of Methionine and Amino Acid Metabolism in Resistant and Induced-Resistant Cells

Metabolomic analyses of cell pellets and culture media were performed for sensitive and resistance cells, before or after induction of resistance. Upon the identification and selection of sensitive and resistant cells, untreated cells underwent metabolic characterization to determine the basal metabolic adaptations that could contribute to resistant mechanisms (e.g., fueling of MDR systems). The results are tabulated in Appendix A. Multivariate analyses of metabolomics data are graphed in Figure 2A–D, including partial least square–discriminant analyses (PLS-DA) and hierarchical clustering analyses of significant metabolites by ANOVA in Ctrl, Res and Ind cells. Res cells were characterized by significantly higher levels of several amino acids, including alanine, arginine, methionine, proline and tyrosine. Similar trends were observed in Ind cells, where the major pathways affected appeared to be methionine and carbon metabolism, with the up-regulation of several intermediates of this pathway, such as choline, methionine, S-Adenosylmethionine (SAM), S-Adenosylhomocysteine (SAH) and glutathione disulfide. Of note, SAM is the major methyl donor for the recovery of oxidized purines (IMP increases in resistant and induced cells compared to control, as does another purine, guanosine—Figure 2B).

This reaction is coupled with the synthesis of polyamines, such as spermidine and spermine, both of which are specifically increased in Res and Ind cells (Figure 3). Additionally, increases in other metabolic intermediates/products of methionine metabolism (acetyl-methionine and methionine sulfoxide, taurine, cysteine) were observed in Res cells, but not in Ind cells (Figure 2B and Figure 3). Since the catabolism of arginine to creatinine depends on SAM, it is interesting to note that reductions in ornithine and citrulline were accompanied by increases in arginine and creatinine in both resistant and induced resistant cells (Figure 3 and Figure 4).

Of all the pathways differentially regulated in both Res and Ind cells, amino acid metabolism ranked among the most significantly altered factors. Therefore, tracking experiments with stable isotope-labeled amino acids in culture were performed to determine whether the accumulation of these amino acids in Res and Ind cells was attributable to increased uptake or decreased consumption for catabolic or anabolic purposes. The results indicate that none of the amino acids were imported more rapidly by resistant or induced resistance cells than in controls (Figure 5). The observation of higher steady state amino acid levels in Res and Ind cells, in the absence of an increased uptake, may potentially be explained by the decrease in mitochondrial amino acid catabolism in resistant cells. The quantification of heavy isotopologues of carboxylic acids from the Krebs cycle indicates a significant increase in succinate and a decrease in malate in resistant (induced) cells compared to controls (Figure 6), indicative of a reduced catabolism and the potential inhibition of complex II of mitochondria at the level of succinate dehydrogenase.

## 3. Discussion

Over time, the search for molecules capable of interfering with cancer cells has resulted in the concomitant observation of cells that survived the treatments. Such results confirm that despite the promising anticancer properties of many compounds, the incomplete eradication of tumor cells is a common process, which is driven by the development of (or positive selection for) clones that are resistant to/tolerant of the anticancer agent. To shed light on the mechanisms driving such resistance/tolerance, we therefore initiated the investigation of surviving cells, and the results reported here clearly show the possibility of inducing vinblastine-resistant murine myeloma cells, and that previously untreated cells were found to be resistant if grown in IM24 medium. 

Drug resistance is a complex process that is influenced by several factors. Here, we show that vinblastine-resistant cells are characterized by metabolic remodeling even prior to exposure to the chemical, which could impact the cells’ ability to counteract drug-induced stress and, possibly, promote a possible epigenetic transition, resulting in “Induced” cells. For example, changes in the extent of methionine uptake and metabolism are suggestive of potential alterations in methylation reactions, redox maintenance, polyamine synthesis and coupling to folate metabolism, thereby coordinating nucleotide and redox status [45,46]. Further studies at the transcription or translational level, including epigenetic investigations, are warranted to follow up on this observation.

One of methionine‘s most important functions is its contribution to intracellular methylation by serving as the sole source of the universal methyl donor S-adenosyl-methionine (SAM); SAM is a necessary substrate for all methylation reactions, including those that modulate gene expression (via methylation of DNA, RNA and histones), phospholipid integrity, the activity of signaling pathways, the damage repair of isoaspartyl proteins and the biosynthesis of polyamines [47].

Of note, lactate was one of the top 10 most increased metabolites in Res cell media compared to Ctrl, suggesting potential glycolytic reprogramming in this group. Since lactate can function as a signaling molecule, it can regulate gene expression through multiple mechanisms, including the lactoylation of histone proteins [48]. 

From metabolomics tracing experiments with stable isotope-labeled amino acids, the elevated steady state levels of methionine in resistant cells seem not to be explained by increased uptake. This result is suggestive of the decreased utilization of intracellular methionine in resistant cells, perhaps as a function of decreased de novo protein synthesis (methionine is the initiator amino acids in all eukaryotic translational processes) or decreased epigenetic regulation via methylation events (of proteins, DNA or RNA). Altogether, the metabolic changes observed could be driven by (i) the law of mass action, as a result of the increased consumption of specific amino acids—perhaps to fuel anaplerotic reactions to sustain viability/proliferation; (ii) the decreased oxidation of specific amino acids at the mitochondrial level, consistent with a metabolic reprogramming towards glycolysis; (iii) epigenetic effects due to a long exposure to the methionine and lactate overproduced by resistance towards induced cells.

This study shows several limitations. First of all, validation in human myeloma cell lines and ensuring translatability into pre-clinical animal models in vivo will be necessary, and is currently underway. Like most single omics studies, tentative interpretations of the data presented here will require orthogonal validation with complementary approaches, such as proteomics and transcriptomics, or more targeted analyses (e.g., expression levels of Pgp for MDR resistance as a function of metabolic adaptations in resistant cells). To determine whether the observed metabolic changes are correlative to or causative of the observed resistant/tolerant phenotypes, experiments aimed at mechanistically testing specific hypotheses (for example, with pharmacological or genetic manipulations) will be needed. However, the findings described in this study represent the first step towards understanding the role of metabolic reprogramming in conferring vinblastine resistance, narrowing the scope of future, more focused investigations.

## 4. Materials and Methods

### 4.1. Cell Growth and Maintenance

Cells from a murine myeloma P3X63Ag8.653 cell line (ATCC, Manasass, VA, USA), derived from the Balb/c strain of mice, were cultured in RPMI-1640 medium supplemented with 10% of heat-inactivated fetal bovine serum and 2 mM glutamine, and incubated at 37 °C in a humidified atmosphere with 5% of CO_2_. Upon reaching confluence, the cells were passed into new culture vessels in a 1:10 ratio and the medium was changed every three days.

### 4.2. MTT Assay

The use of the activity of the mitochondrial enzyme succinate dehydrogenase as a detector of cell viability is possible through its ability to reduce the tetrazolium dye MTT, or 3-(4,5-dimethylthiazol-2-yl)-2,5-diphenyltetrazolium bromide, into its insoluble formazan crystals. Since a reduction in MTT is correlated with cell metabolic activity, metabolically active cells are expected to yield high MTT values.

The MTT assay was performed to evaluate the effect of vinblastine on cell viability [43]. In total, 2 × 10^4^ cells/well in 100 µL of complete RPMI medium were seeded in a 96-well micro plate, and after 24 h of incubation different concentrations of vinblastine (from 1 μM to 0.0009 μM, twice to ten-times diluted) were added. After 24 h, the medium containing the treatment was removed and 100 μL of MTT solution, at the final concentration of 0.5 mg/mL, was added to each well and incubated in the dark at 37 °C for 3 h. The formazan crystals produced were dissolved in 100 μL of DMSO and the optical density (OD) measured at 595 nm by a Tecan Sunrise™ (Tecan Group Ltd., Männedorf, Switzerland) UV-Vis spectrophotometer. The values were expressed as a percentage of cell viability, obtained using the following equation: Cell viability (%) = (OD treated sample / OD untreated sample) × 100

The percent cell viability data, obtained by converting the optical density values as previously described, were processed with an AAT Bioquest EC_50_ Calculator (Sunnyvale, CA, USA) [49] to obtain the concentration at which the vinblastine exerts half of its maximum response values (EC_50_). The values were repeated three times and are reported as mean ± SD.

### 4.3. Resistant and Induced Cells

In order to induce drug resistance to vinblastine, murine myeloma cells were seeded at a concentration of 1.5 × 10^4^ cells/mL in a six-well plate and treated for 7 days with low doses of vinblastine (vinblastine sulfate salt ≥ 97% HPLC purity grade, obtained by SIGMA Aldrich—product no: V1377), and 5 nM equal to 1/4 of the of EC_50_ value against P3X63Ag8.653 cells (20 nM). At the end of the treatment period, 1.5 × 10^4^ Res cells/mL were collected and seeded in a six-well plate with fresh culture medium to obtain an Induction Medium (IM24). After 24 h, the IM24 was collected and centrifuged twice at 1500 rpm for 5 min at room temperature. After discarding the pellet (consisting of Res and cell debris), the resulting medium was used to culture P3X63Ag8.653 cells, and after 24 h Ind cells were obtained. P3X63Ag8.653 cells cultured under the same conditions as Res and Ind cells were used as control (Ctrl Res and Ctrl Ind, respectively).

To evaluate resistance to vinblastine 2 × 10^4^ cells/well of Ctrl, Ctrl Res, Ctrl Ind, Res and Ind were seeded in a 96-well plate, treated for 24 h with 20 nM vinblastine and subjected to MTT assay to evaluate the percentage of cell viability. 

### 4.4. Ultra-High-Pressure Liquid Chromatography–Mass Spectrometry (MS) Metabolomics and Tracing Experiments

Approximately 2 × 10^6^ cells were collected and extracted in 1000 µL of ice cold extraction solution (methanol:acetonitrile:water 5:3:2 *v*/*v*/*v*). Suspensions were vortexed continuously for 30 min at 4 °C. Insoluble material was removed by centrifugation at 18,000× *g* for 10 min at 4 °C, and supernatants were isolated for metabolomics analysis by UHPLC-MS. Media samples (20 µL) were extracted as described above, in a ratio of 1:25 with the extraction solution.

Analyses were performed using a Vanquish UHPLC coupled online to a Q Exactive mass spectrometer (Thermo Fisher, Bremen, Germany). Ten microliters of sample extracts were loaded onto a Kinetex XB-C18 column (150 × 2.1 mm i.d., 1.7µm—Phenomenex). The samples were analyzed using the 3 min isocratic condition or a 5, 9, and 17 min gradient, as described [50,51,52]. 

Metabolite assignments, amino acids-labeled tracing experiments, isotopologue distributions, and correction for expected natural abundances of ^13^C and ^15^N isotopes were performed using MAVEN (Princeton, NJ, USA) [53].

### 4.5. Statistical Analyses

Data were expressed as mean ± standard deviation (SD) from three independent experiments. Statistical significance between examined samples and control values was determined using one-way analysis of variance. Graphs and statistical analyses (either PLSDA, HCA or repeated measures ANOVA) were processed with GENE-E (Broad Institute, Cambridge, MA, USA), and MetaboAnalyst 5.0 [54]. 

## 5. Conclusions

Over time, in our search for molecules capable of interfering with cancer cells, we observed cells surviving drug treatment, thus confirming the development of (or positive selection for) clones that are resistant to/tolerant of the anticancer agent. The phenomenon of drug resistance gives cancer resistance and tolerance to anticancer agents. To improve the knowledge about drug resistance linked to the use of vinblastine, a natural product widely used in clinical applications for cancer therapy, as a first step we conducted investigations on murine myeloma cells. We obtained vinblastine-resistant cells and observed the ability of their culture medium to induce vinblastine resistance in myeloma cells never before cultured in the presence of vinblastine. Further, the metabolomic analysis clearly showed that some metabolites produced higher amounts in both resistant and induced cells, and confirmed a process of reprogramming the metabolism of murine myeloma cells kept in culture in the presence of vinblastine (resistant cells), and in murine myeloma cells not treated with vinblastine and cultured in a culture medium taken from resistant cells. Our results will be useful for further investigations of human cancer cells.

## Figures and Tables

**Figure 1 molecules-28-02051-f001:**
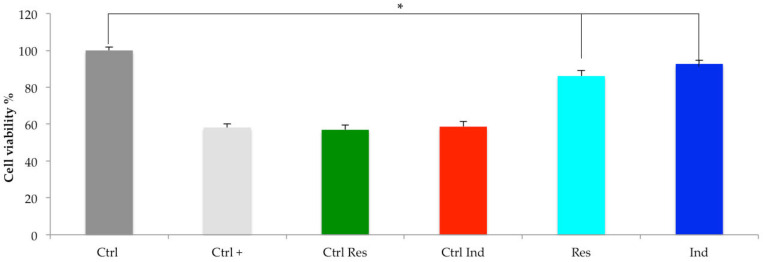
Bar graph of cell viability % of P3X63Ag8.53 cells treated with EC50 values of vinblastine (20 nM). In dark gray, the bar relating to normal untreated cells (Ctrl); in light gray the normal cells treated with 20 nM of vinblastine (Ctrl+); in green the normal cells grown under the same culture conditions as Res cells, but without a lower-dosage vinblastine treatment (Ctrl Res); in red the cells grown for 24 h in Ctrl Res cell culture medium or IM24 (Ctrl Ind); in greenish-blue color the resistant cells treated for seven days with low doses of vinblastine (Res); in blue the induced cells cultured for 24 h in the cell culture medium Res (Ind). * *p* < 0.05 significant differences compared with Ctrl+, Ctrl Ind and Ctrl Res.

**Figure 2 molecules-28-02051-f002:**
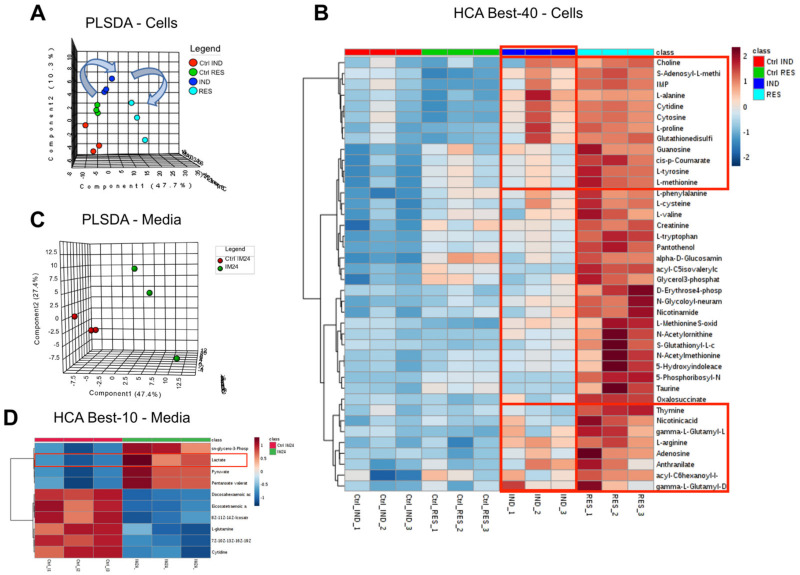
Multivariate analyses of metabolomics data from cell extracts and tissue cultures in control, Res and Ind cells. These analyses include partial least square discriminant analyses (PLS-DA) of cell extracts (**A**), hierarchical clustering analysis and a heatmap of the top 40 metabolites by ANOVA (**B**), and PLS-DA and a heatmap (top 10) of the media ((**C**,**D**), respectively).

**Figure 3 molecules-28-02051-f003:**
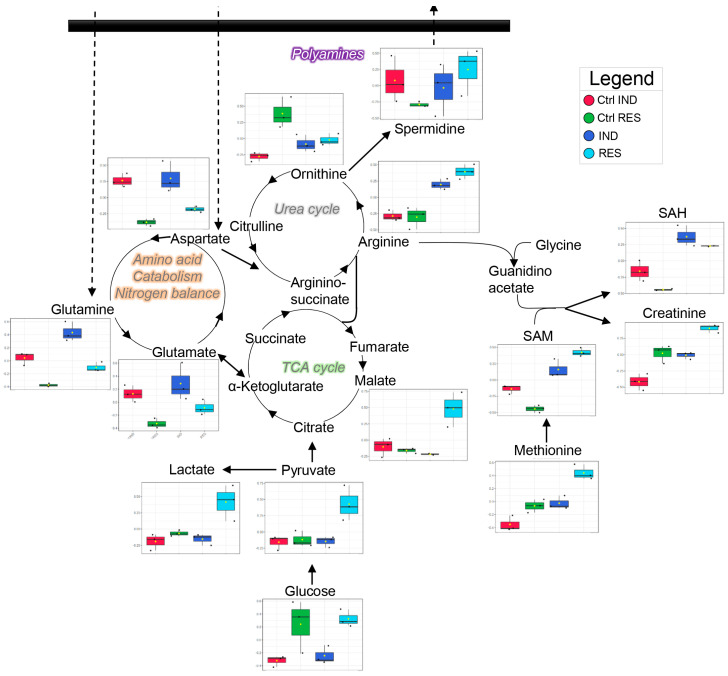
An overview of the metabolism of arginine and methionine in the cell extracts of the four groups tested in this study (color legend in the top right corner of the figure).

**Figure 4 molecules-28-02051-f004:**
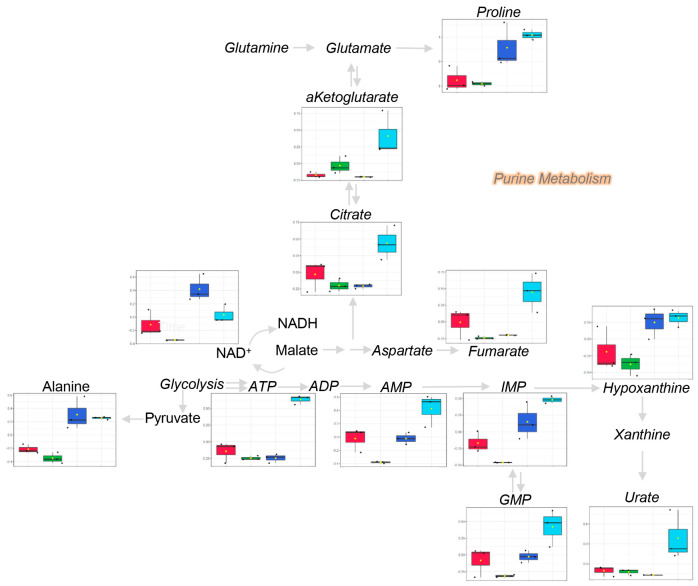
An overview of arginine and methionine metabolism in cell extracts from the four groups tested in this study (color codes consistent with Figure 3).

**Figure 5 molecules-28-02051-f005:**
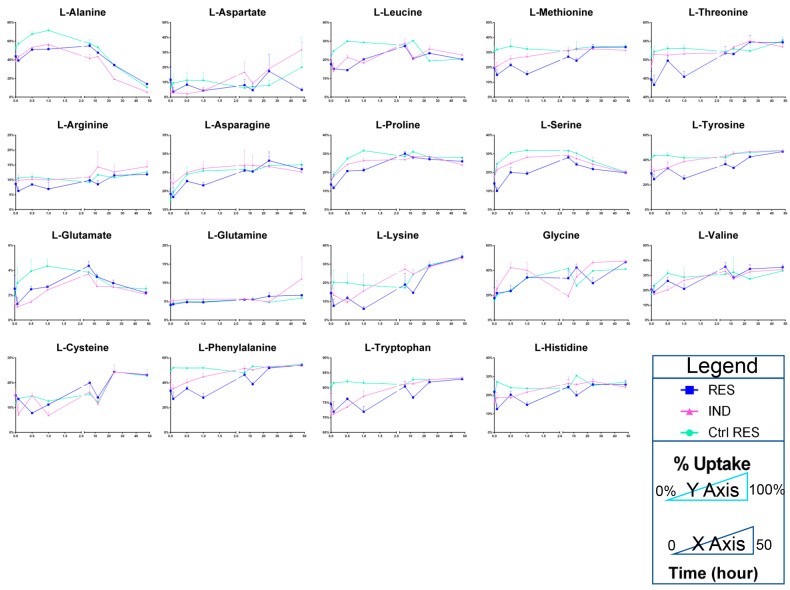
Res cells, Ind and Ctrl Res cells (color legend in the lower right corner of the figure) were incubated with a cocktail of ^13^C ^15^N-labeled amino acids, to determine amino acid uptake rates over a range of time of 50 h (*x* axis). Percent of amino acid uptake is shown on the *y* axis, determined as % labeled isotopologues of each amino acid relative to the total levels (including unlabeled endogenous ones).

**Figure 6 molecules-28-02051-f006:**
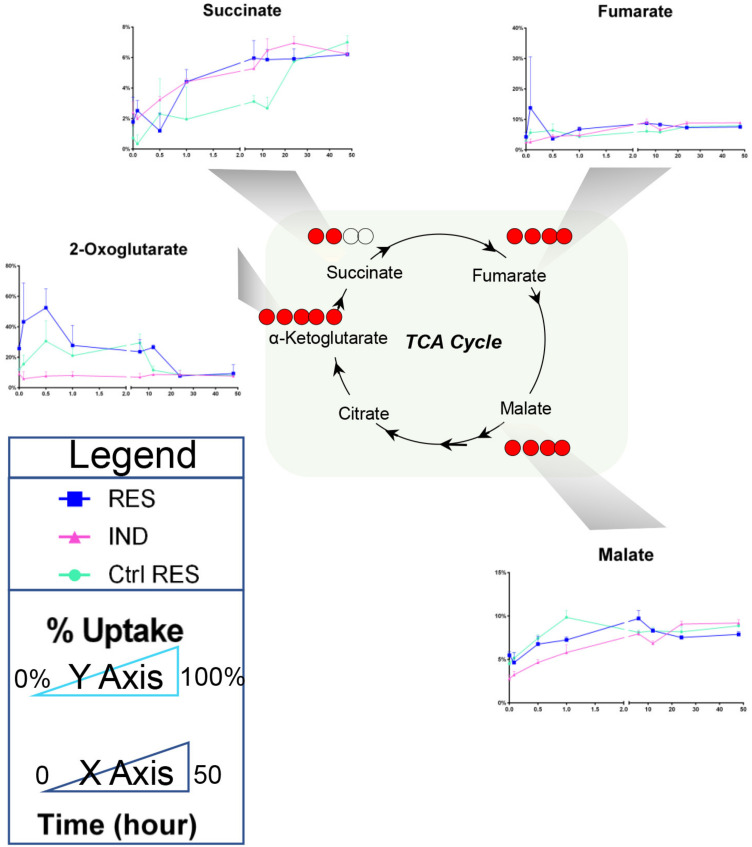
Percent accumulation of Krebs cycle labeling of resistant cells, induced resistant cells and control cells (color legend in the lower left corner of the figure), after incubation with a mix of ^13^C ^15^N-labeled amino acids, to determine the catabolism of amino acids over a time interval of 50 h (x axis). The percentage of labeled carboxylic acids is shown on the y axis, determined as the % of labeled isotopologues of each intermediate of the Krebs cycle relative to the total levels (including unlabeled ones, as derived from the catabolism of other substrates).

## Data Availability

The P3X63Ag8.653 cell line was purchased by ATCC (Manasass, VA, USA); the RES and IND cells were obtained in our laboratory by our experimental work.

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
