# Peer review of "Induction of Drug-Resistance and Production of a Culture Medium Able to Induce Drug-Resistance in Vinblastine Untreated Murine Myeloma Cells"

_molecules, 2023, doi:10.3390/molecules28052051_

Round 1

Reviewer 1 Report

In this manuscript, Valentino et al. reported on Induction of Vinblastine Drug Resistance in Murine Myeloma Cells. Considering the novelty and utility of this manuscript, it can be accepted in Molecules. This is a good quality manuscript and there is an improvement, if the authors can examine the variations in Genes’ expressions in tumor cells. Also more literatures survey is needed.

Author Response

In this manuscript, Valentino et al. reported on Induction of Vinblastine Drug Resistance in Murine Myeloma Cells. Considering the novelty and utility of this manuscript, it can be accepted in Molecules. This is a good quality manuscript and there is an improvement, if the authors can examine the variations in Genes’ expressions in tumor cells. Also more literatures survey is needed.

We thank the referee for the suggestion that we will follow in the next steps of the future research. As requested, we added new references in order to improve the paper. We are currently planning proteomics investigations of the three cell lines, though current funding limitations constraint the feasibility of additional studies prior to the publication of the present manuscript.

Reviewer 2 Report

In the current manuscript Valentina et al. describe a metabolomic study on murine myeloma cells with acquired vinblastine resistance. While drug resistance is one of the most important topics in cancer research and metabolomic studies open up new avenues to understand it even more, this work fell short in several ways to be published in Molecules.

1. How and why was the murine myeloma P3X-Ag8.653 cell line chosen for this study? What is the relevance? Vinblastine is not used routinely in myeloma treatment. Moreover, why was a murine cell line selected over a human cell line? And even if the cell line would be appropriate for this work, what kind of conclusions could be drawn from only one cell line?

2. Why was vinblastine-sensitivity (or resistance) evaluated at only one concentration on Fig.1? If a complete cytotoxicity curve was generated, please include those for all cell lines, not just one condition! The shift in the IC50 values will show the rate of drug resistance!

3. Based on the presented figures the induction of resistance in the P3X-Ag8.653 cells are not convincing at all. Showing only a slight change in vinblastine sensitivity at only one concentration is not enough. Robust and reproducible results are required to prove that a 7-day treatment induce significant resistance in these cells.

4. The authors themselves state in the Introduction that the main contributor in vinblastine resistance is the overexpression of MDR1 (Abcb1, P-glycoprotein), however they didn’t check whether the expression or activity of MDR1 is increased after drug selection. This could explain the slightly increased tolerance to vinblastine. Actually, no other known drug resistance mechanism was investigated in this work.

5. I understand this metabolomic approach could be interesting for characterizing (drug resistant) cancer cells, however in this manuscript I didn’t find a clear explanation why it’s good, why it’s important and how it helps to investigate drug resistance while no true mechanisms of drug resistance were checked. On the contrary, I’ve found it unnecessary and over-explained. Vinblastine is a microtubule assembly inhibitor, what could be the possible explanation of vinblastine-resistance through these suggested metabolomic changes? Is there any logical link between these two?

6. The cells were treated with vinblastine for 7 days than a viability assay and the metabolomic analysis were done, if I understood it correctly. If it’s true than no stable “vinblastine resistant” (if we accept that slight change in viability as drug resistance…) cells were used, only cells which were currently taken out from vinblastine treatment. Therefore, the observed metabolomic changes are attributed the vinblastine damage itself rather than an emerging vinblastine resistance. Drug resistant cell lines usually considered drug resistant when the resistance (at least 2-fold, statistically significant IC50 increase) is proven and it’s stable for several passages.

In summary, the study design, the experimental model and the conclusions are below the standards of the journal Molecules, therefore I cannot suggest it for publication in it’s current form.

Author Response

In the current manuscript Valentina et al. describe a metabolomic study on murine myeloma cells with acquired vinblastine resistance. While drug resistance is one of the most important topics in cancer research and metabolomic studies open up new avenues to understand it even more, this work fell short in several ways to be published in Molecules.

  1. How and why was the murine myeloma P3X-Ag8.653 cell line chosen for this study? What is the relevance? Vinblastine is not used routinely in myeloma treatment. Moreover, why was a murine cell line selected over a human cell line? And even if the cell line would be appropriate for this work, what kind of conclusions could be drawn from only one cell line?

Thank you for the very pertinent and constructive comments. While vinblastine is now a mainstay in chemotherapy of some blood cancers (e.g., Hodgkin’s lymphoma), it was originally tested in clinical trials focusing on multiple myeloma (Cancer Chemother Rep . 1963 Mar;27:87-9.). This is now clearly stated in the manuscript, as it should have in the first place. In keeping with the previous statement, here we focused on a myeloma cell line. While we totally appreciate and second the reviewer’s comment on the higher level of translatability of translational studies on human cell lines, the choice to use the P3X-Ag8.653 cell line was dictated by ease of availability in our laboratory and, as indicated by commercial vendor, by the appreciation that this cell line is “Comparable to other myeloma lines used for fusions relating to growth properties, fusion frequency and stability of resulting hybrids” (https://www.sigmaaldrich.com/US/en/product/sigma/cb_85011420 ). However, given the interesting results generated as part of this study, we will pursue further similar investigations on human cell lines (presently running in our Lab). These considerations are now included in the revised version of the manuscript.

  1. Why was vinblastine-sensitivity (or resistance) evaluated at only one concentration on Fig.1? If a complete cytotoxicity curve was generated, please include those for all cell lines, not just one condition! The shift in the IC50 values will show the rate of drug resistance!

The complete cytotoxicity curve and the rate of drug resistance were not performed for all cell lines. The dose used here was selected based on the EC50 for vinblastine for Ctrl cells (P3X-Ag8.653).

  1. Based on the presented figures the induction of resistance in the P3X-Ag8.653 cells are not convincing at all. Showing only a slight change in vinblastine sensitivity at only one concentration is not enough. Robust and reproducible results are required to prove that a 7-day treatment induce significant resistance in these cells.

In our Lab the experimental procedure here reported has been repeated many times and the same results were obtained and validate by statistical analysis (ANOVA).

  1. The authors themselves state in the Introduction that the main contributor in vinblastine resistance is the overexpression of MDR1 (Abcb1, P-glycoprotein), however they didn’t check whether the expression or activity of MDR1 is increased after drug selection. This could explain the slightly increased tolerance to vinblastine. Actually, no other known drug resistance mechanism was investigated in this work.

We agree with referee comments. In our experimental planning, we evaluated such a possibility and the overexpression of MDR1 will be tested in further investigations. In this paper we focussed our attention on the finding that the culture medium of resistant cells was able to induce resistance to vinblastine in untreated cells.

  1. I understand this metabolomic approach could be interesting for characterizing (drug resistant) cancer cells, however in this manuscript I didn’t find a clear explanation why it’s good, why it’s important and how it helps to investigate drug resistance while no true mechanisms of drug resistance were checked. On the contrary, I’ve found it unnecessary and over-explained. Vinblastine is a microtubule assembly inhibitor, what could be the possible explanation of vinblastine-resistance through these suggested metabolomic changes? Is there any logical link between these two?

Clearly, the reviewer was right when pointing out that we totally left the rationale for performing metabolomics analyses in the context of vinblastine-resistance unexplained or, at least, dangling – given the mention of the MDR system. The introduction has been revised to clarify this point.

  1. The cells were treated with vinblastine for 7 days than a viability assay and the metabolomic analysis were done, if I understood it correctly. If it’s true than no stable “vinblastine resistant” (if we accept that slight change in viability as drug resistance…) cells were used, only cells which were currently taken out from vinblastine treatment. Therefore, the observed metabolomic changes are attributed the vinblastine damage itself rather than an emerging vinblastine resistance. Drug resistant cell lines usually considered drug resistant when the resistance (at least 2-fold, statistically significant IC50 increase) is proven and it’s stable for several passages.

The induced cells did not come in contact with vinblastine before being processed for metabolomics investigations, only with the culture medium obtained from 24h cultured resistant cells. Consequently, the metabolic changes observed in the induced cells are not vinblastine-dependent. The rationale behind this experiment was to determine whether basal metabolic alterations could be observed in cells that would be resistant to vinblastine. This is now clarified in the revised version of the manuscript.

Reviewer 3 Report

Authors conducted elucidation of drug resistance of vinblastine. Drug resistance are probably complicated manners. Therefore, these are important results for elucidation of drug resistance. I recommend this manuscript is to be published in Molecules.

1. Authors should describe compony and purity of vinblastine.

2. If possible, add chromatograms of LC-MS in Supplementary Materials.

Author Response

  1. Authors should describe compony and purity of vinblastine.

Vinblastine sulphate salt ≥97% (HPLC), obtained by SIGMA Aldrich (product no: V1377)

  1. If possible, add chromatograms of LC-MS in Supplementary Materials.

Since all metabolites were determined via a semi-targeted LC-HRMS analysis (untargeted intact mass data acquisition, manual peak picking via mining of the Extract Ion Chromatograms through Maven) it would be rather unusual for us to include all the single chromatograms in supplementary materials. However, we are glad to share the .raw or .mzxml files used to generate the data to any interested reader upon request.